# Benefits of VR Physical Exercise on Cognition in Older Adults with and without Mild Cognitive Decline: A Systematic Review of Randomized Controlled Trials

**DOI:** 10.3390/healthcare9070883

**Published:** 2021-07-13

**Authors:** Kohei Sakaki, Rui Nouchi, Yutaka Matsuzaki, Toshiki Saito, Jérôme Dinet, Ryuta Kawashima

**Affiliations:** 1Department of Functional Brain Imaging, Institute of Development, Aging and Cancer, Tohoku University, Sendai 980-8575, Japan; toshiki.saito.e1@tohoku.ac.jp (T.S.); ryuta@tohoku.ac.jp (R.K.); 2Department of Cognitive Health Science, Institute of Development, Aging and Cancer, Tohoku University, Sendai 980-8575, Japan; rui@tohoku.ac.jp; 3Division of Developmental Cognitive Neuroscience, Institute of Development, Aging and Cancer, Tohoku University, Sendai 980-8575, Japan; yutaka.matsuzaki.e5@tohoku.ac.jp; 4Psychology and Neuroscience Laboratory, Université de Lorraine, 2LPN, F-54000 Nancy, France; jerome.dinet@univ-lorraine.fr

**Keywords:** virtual reality, cognitive function, cognitive declines, physical intervention, older adults

## Abstract

It is well known that physical exercise has beneficial effects on cognitive function in older adults. Recently, several physical exercise programs with virtual reality (VR) have been proposed to support physical exercise benefits. However, it is still unclear whether VR physical exercise (VR-PE) has positive effects on cognitive function in older adults. The purpose of this study was to conduct a systematic review (SR) of the effects of VR-PE on cognitive function in older adults with and without cognitive decline. We used academic databases to search for research papers. The criteria were intervention study using any VR-PE, participants were older adults with and without mild cognitive decline (not dementia), and cognitive functions were assessed. We found that 6 of 11 eligible studies reported the significant benefits of the VR-PE on a wide range of cognitive functions in aging populations. The SR revealed that VR-PE has beneficial effects on the inhibition of executive functions in older adults with and without mild cognitive decline. Moreover, VR-PE selectively leads to improvements in shifting and general cognitive performance in healthy older adults. The SR suggests that VR-PE could be a successful approach to improve cognitive function in older adults with and without cognitive decline.

## 1. Introduction

Cognitive function declines with age [1]. This decline is an indicator of lower well-being [2] and a risk of dementia in the future [3,4]. Considering the growth of the older adult population, it is important to delay or prevent dementia in healthy older adults and older adults with mild cognitive decline [5]. Physical exercise (PE) plays an important role in maintaining and improving cognitive function in the aging population [6]. A previous cohort study has reported that physical exercise reduces dementia risk in older adults [7]. Previous intervention studies have reported that physical exercise programs using aerobic exercise [8,9], balance exercise [9], combination physical exercises [10,11], or cardiovascular and coordination training [12] have acute and long-term benefits on cognitive functions.

Although these previous studies have demonstrated the beneficial effects of physical exercise on cognitive functions, a large proportion of the aging population does not participate in adequate physical exercise [13]. There are external and internal barriers for older adults to participate in physical exercise [14]. For example, older adults have reported that the “lack of infrastructure” is one of the main external barriers and “lack of motivation” one of the main internal barriers [14].

Recent technologies, such as the virtual reality (VR) technique, are expected to support physical exercise in the aging population. VR is defined as the simulation, in real time, of an interactable environment, scenario, or activity [15]. Several VR physical exercises (VR-PE) have been proposed [16,17,18,19,20,21,22,23,24,25,26], in which people at their homes use commercially available VR systems, such as video gaming consoles or personal computers [15]. These products can help to overcome the barriers of lack of infrastructure and motivation since almost all VR-PE include gaming factors to enhance motivation and increase user participation [27]. Previous studies have shown that physical exercise with VR has the potential to increase exercise behavior in older adults [28].

Previous intervention studies using gaming factors showed significant improvements in a wide range of cognitive functions and a small dropout ratio from the intervention in older adults [29,30,31,32]. It is also reported that VR-PE led to improvements both in cognitive function in healthy older adults [17,21,25,26] and older adults with cognitive decline [16,22]. Therefore, it seems possible to use VR-PE to enhance cognitive function in older adults. However, to the best of our knowledge, there is no systematic review (SR) and meta-analysis of RCTs to assess the benefits of VR-PE on cognitive function in older adults with or without mild cognitive decline (not dementia). Therefore, we aimed to conduct a systematic review of the beneficial effects of VR-PE on cognition.

## 2. Materials and Methods

### 2.1. Protocol and Registration

This protocol followed the statement and general principles of Preferred Reporting Items for Systematic Reviews and Meta-Analyses (PRISMA) statement [33] (Appendix A), and it was designed using the International Prospective Register of Systematic Reviews (PROSPERO) with the registration number CRD42020220020 (https://www.crd.york.ac.uk/prospero/display_record.php?RecordID=220020 (accessed on 10 December 2020)).

### 2.2. Search Strategy

Our review question was “does VR-PE have the effect of improving cognitive functions in older adults with and without mild cognitive declines?”. We searched PubMed and Scopus for studies using specific search terms (Appendix A). Only articles published in English were included, and their publication period was unrestricted.

### 2.3. Inclusion and Exclusion Criteria

This SR included studies where participants were men and women aged 60 years or older with and without mild cognitive decline. Participants with a current diagnosis of Alzheimer’s disease, dementia, vascular dementia, stroke, head injury, depression, or other neurologic disorders were excluded. Only intervention studies to assess the beneficial effects of VR-PE on cognitive function were included. We defined VR as the simulation, in real time, of an interactable environment, scenario, or activity [15]. There were no restrictions on which type of VR device could be used for intervention. The eligible outcomes were cognitive functions measured using any validated measure, including computerized tests. Cognitive functions, including general cognitive functions, executive function, working memory capacity/short-term memory, episodic (long-term) memory, processing speed, and attention, among others, were considered in this SR.

### 2.4. Quality Assessment

The quality of each study was assessed using the modified Delphi list [34]. These quality assessment criteria were based on previous systematic review papers [35,36].

## 3. Results

### 3.1. Study Selection and Characteristics

Overall, 412 scientific articles were identified through database searches. After excluding duplicates (*n* = 36), we identified 376 articles. At the title and abstract screening, we excluded 352 studies that were not intervention studies, did not assess cognitive functions as an outcome, or intended to treat physical, cognitive, or mental diseases. Of the 24 articles selected for full-text assessment, 13 were excluded for including clinical patients who suffered from physical or cognitive disorders (*n* = 4) or young adults (*n* = 2), case reports (*n* = 2), secondary analysis (*n* = 2), one-shot intervention (*n* = 1), no cognitive assessments (*n* = 1), or full-text unavailable (*n* = 1). Eleven articles were eligible for the current review, based on the inclusion and exclusion criteria. The study selection process is presented in the PRISMA flowchart (Figure 1). The characteristics of the included studies are summarized in Table 1. Six studies included healthy older adults [16,18,20,22,25,26], and five included older adults with mild cognitive decline (mild cognitive impairment [MCI] or mild dementia) [17,19,21,23,24]. The sample size ranged from 10 to 84. The mean age of the participants ranged from 68.0–87.2 years.

### 3.2. Quality Assessment

The methodological quality of the included studies is shown in Table 2. The range of the quality assessment score was 6–12, with an average of 8.1 (SD = 2.02).

Eight studies fulfilled the requirements of Item 11 (adequate description of the control/comparison group) [16,17,18,20,22,24,25,26]. Seven of eight studies used the physical exercise only group as the control group [16,17,18,20,22,25,26]. Two studies used the physical and cognitive exercise group as the control group [22,24]. The other three studies used a no-intervention group as the control group [19,21,23].

The scores of Item 2 (Treatment Allocation Concealed), Item 5 (Blinded Outcome Assessor), Item 6 (Care Provider Blinded), Item 7 (Patient Blinded), Item 9 (intention-to-treat analysis), Item 10 (details of random allocation methods), and Item 14 (reporting CONSORT statement) were low. It seems to be difficult to blind participants and caregivers on the type of intervention they received because of the nature of the intervention methods. Thus, only five studies used blinded outcome assessors.

### 3.3. Intervention

The characteristics of intervention methods in the included studies are summarized in Table 3. The intervention period of the included studies was six-weeks–six-months. Three studies were conducted for six months [17,20,21], two studies were conducted for three months [16,24], one study was conducted for eight weeks [22], one study was conducted for seven weeks [18], and four studies were conducted for six weeks [19,23,25,26].

Three studies used a stationary bike equipped with a VR display (“Cybercycle” or “Cycle-ergometer”) as the method of VR-PE [16,17,23]. One study used Kayak ergometers with 3-D images on the screen [26]. Five studies used a motion capture system (“Xbox 360 Kinect” or “Wii Fit”) to provide feedback information on movements during physical exercise [18,21,22,24,25]. Two studies used a pressure sensitive platform (“BioRescue” or “Impact Dance Platforms”) to provide feedback information about steps or balance during physical exercise [19,20].

### 3.4. Outcome

Among the 11 studies, a variety of cognitive tests were used to examine the impact of the intervention on cognitive function. Some studies employed a general assessment measure, such as the Mini-Mental State Examination (MMSE) or Montreal Cognitive Assessment (MoCA). Others combined multiple tests to assess a single cognitive domain. In this SR, the cognitive tests were divided into seven cognitive domains—general cognitive functions, executive function, working memory capacity/short-term memory, episodic (long-term) memory, processing speed, attention, and others—based on a previous study [35]. The various psychological tests used to assess cognitive function in the included studies are summarized in Table 4.

#### 3.4.1. General Cognitive Functions

Two general cognitive function outcomes were measured using the MMSE or MoCA in seven studies [18,19,22,23,24,25,26]. The test of general cognitive functions performed for healthy older adults indicated a statistically significant improvement in MoCA compared to the control group [22,26]. Another study on healthy older adults [18] and older adults with mild cognitive decline [24] showed a statistically significant improvement in MoCA from baseline. In contrast, one study of healthy older adults and two studies of older adults with mild cognitive decline reported no significant change in the MoCA score [19] or MMSE score [23,25].

#### 3.4.2. Executive Function

Four cognitive domains (inhibition, shifting, updating, and the others) in executive function assessed by 12 tests (trail making test part B (TMT-B) [20,23], color trails [16,17], Stroop [16,17], verbal fluency [16,23,25], executive control task [20], timed up and go test cognition (TUG-cog) [22], abstract thinking and judgment [21], animal name fluency [21], frontal assessment battery (FAB) [23], TUG dual attention task (TUG-DT) [19], executive interview 25 (EXIT-25) [24], floor maze test (FMT) [25]) were assessed in nine studies.

Inhibition measured by the Stroop test performed by healthy older adults indicated a statistically significant improvement compared to the control group [16]. One study of older adults with mild cognitive decline showed statistically significant improvements in the Stroop test from baseline [23].

Shifting as measured by color trails performed by healthy older adults indicated statistically significant improvements compared to the control group [16] but not in older adults with mild cognitive decline [17]. TMT-B performed by healthy older adults indicated a statistically significant improvement from baseline [20]. One study that performed TMT-B in older adults was excluded from the analysis because the number of participants who completed the task was too small [23].

Updating measured by the executive control task performed for healthy older adults indicated a statistically significant improvement from the baseline [20]. Other studies performing verbal fluency or animal name fluency in healthy older adults [25] and in older adults with mild cognitive decline [21,23] did not indicate statistically significant improvements.

Other types of executive function measures were used in six studies [19,21,22,23,24,25]. Dual-task performance during motor and cognitive tasks measured by TUG-cog performed by healthy older adults indicated statistically significant improvement compared to the control group [21] but not TUG-DT performed by older adults with mild cognitive decline [19]. Abstract thinking in the cognitive abilities screening instrument (CASI) [37] performed by older adults with mild cognitive decline indicated statistically significant improvement compared to the control group [21]. FAB performed by older adults with mild cognitive decline did not indicate a statistically significant improvement [23]. EXIT-25 performed by older adults with mild cognitive decline indicated statistically significant improvements from the baseline [24]. The floor maze test performed by healthy older adults did not indicate a statistically significant improvement [25].

#### 3.4.3. Working Memory Capacity and Short-Term Memory

Working memory capacity was assessed using the backward digit Span in four studies [16,17,20,25]. One study that performed digit span backward in healthy older adults indicated statistically significant improvements compared to the control group [16]. Additionally, one study performed in healthy older adults indicated statistically significant improvements from the baseline [20]. In contrast, one study performed in healthy older adults [25] and in older adults with mild cognitive decline [17] reported no significant change.

Two short-term memory outcomes were assessed using the digit span forward or the subtest of CASI in three studies [20,21,25]. One study that performed digit span forward in healthy older adults indicated statistically significant improvements compared to the control group [25]. In contrast, one study performed in healthy older adults [20] and in older adults with mild cognitive decline [21] reported no significant change.

#### 3.4.4. Episodic (Long-Term) Memory

Verbal episode memory outcomes were assessed using the Rey auditory verbal learning test (RAVLT) [16,23], ADAS word list [17], logical memory (story recall) in WMS-R [20], and Chinese version of the California verbal learning test (CVVLT) [24] in five studies. One study that performed the ADAS word list for older adults with mild cognitive decline indicated statistically significant improvement compared to the control group [17]. Additionally, one study performing logical memory (story recall) in WMS-R for healthy older adults [20] and one study performing CVVLT for older adults with mild cognitive decline [24] indicated a statistically significant improvement from the baseline but not RAVLT [16,23].

Visual episode memory outcomes were assessed using the paired associates learning task [20] or delayed recall [16]. The paired associates learning task performed for healthy older adults indicated statistically significant improvements from the baseline [20] but not in figure delayed recall [16].

#### 3.4.5. Processing Speed

Two processing speed outcomes were assessed using the trail making test part A (TMT-A) and digit symbol substitution task from Wechsler adult intelligence scale-revised (WAIS-R) in three studies [20,23,25]. The tests of processing speed performed by healthy older adults indicated statistically significant improvements from the baseline [20]. In contrast, one study performed in healthy older adults [25] and in older adults with mild cognitive decline [23] reported no significant change.

#### 3.4.6. Attention

Four attention outcomes were assessed using a subtest of CASI (ATTEN), attentional matrices test, age concentration tests A and B, and letter digit symbol test in four studies [16,20,21,23]. Only age concentration tests A and B performed by healthy older adults indicated statistically significant improvements from the baseline [20], but this did not occur in three studies [16,21,23].

#### 3.4.7. Others

Visuospatial functions were assessed in two studies [16,23]. Figure copy and clock performed by healthy older adults [16] and the Rey-Osterrieth complex figure test performed by older adults with mild cognitive decline [23] did not indicate statistically significant improvements.

The ecological validity questionnaire, which is a self-report measure of cognitive function, was assessed in a study of older adults with mild cognitive decline and showed statistically significant improvements compared to the control group [17].

Subdomains in the CASI (mental manipulation, orientation, language, drawing, long-term memory) performed by older adults with mild cognitive decline did not indicate statistically significant improvements [21].

## 4. Discussion

This study first conducted an SR to investigate the effects of VR-PE on cognitive functions in older adults with and without mild cognitive decline. Eleven studies (six studies including healthy older adults and five studies including older adults with mild cognitive decline) met the SR criteria [16,17,18,19,20,21,22,23,24,25,26]. We found that 6 of 11 studies (four studies for healthy older adults and two studies for older adults with mild cognitive decline) reported the significant benefits of VR-PE on general cognitive function [22,26], executive function [16,21,22], working memory capacity [16], short-term memory [25] and verbal episodic memory [17] compared to the control groups.

In the SR, three of nine studies that assessed executive functions reported a significant improvement in executive functions in healthy older adults and older adults with mild cognitive decline compared to the control groups. These studies included diverse aspects of executive function, such as inhibition, shifting, updating, dual-task performance, and abstract thinking in older adults with and without mild cognitive decline. We discuss the effects of VR-PE on each subdomain in executive functions separately. For inhibition, VR-PE should have beneficial effects on inhibition performance in older adults with and without mild cognitive decline [16,23]. For shifting, results indicate a potential beneficial effect of VR-PE on shifting performance as measured by trial making [20] or color trials [16] in healthy older adults. For the dual-task task, we found only two studies that measured dual-task performance [19,22] and one study that measured abstract thinking [21]. It is difficult to conclude the positive effects of VR-PE on dual-task and abstract thinking in executive functions. In future studies, it would be important to investigate the beneficial effects of VR-PE on dual-task and abstract thinking in older adults.

We found inconsistent results in general cognitive function, working memory capacity, short-term memory, and verbal episodic memory. For general cognitive function, two of seven studies indicate statistically significant improvement compared to the control group [22,26] but not the five studies [18,19,23,24,25]. For working memory capacity and short-term memory, the result should be interpreted with limitations due to the small number of studies that assessed working memory capacity and short-term memory [20,21,25]. For verbal episodic memory, results indicate that VR-PE does not have positive effects on verbal memory performance in older adults with and without mild cognitive decline [16,17,20,23,24].

It is important to consider the differences in the effects of VR devices on cognition among the included studies. Four types of VR devices were included in the studied papers: stationary bikes equipped with VR displays (“Cybercycle” or “Cycle-ergometer”), Kayak ergometers, motion capture (“Xbox 360 Kinect” or “Wii Fit”), and pressure-sensitive platforms (“BioRescue” or “Impact Dance Platforms”). We did not find any significant improvements when participants used pressure-sensitive platforms compared to the control group [19,20]. However, for motion capture, three of five studies reported significant improvements in abstract thinking and dual task performance in executive functions in healthy older adults [22] as well as older adults with mild cognitive decline [21] and in short-term memory in healthy older adults [25]. For the stational bike equipped with VR, one of two studies showed significant improvements in shifting and inhibition performance in executive functions in healthy older adults [16]. Moreover, one study reported a pre-post difference in inhibition in older adults with mild cognitive decline [17]. For the Kayak ergometers, one study showed significant improvements in general cognitive function in healthy older adults [26]. Taken together, the motion capture, stationary bike equipped with a VR display, and Kayak ergometers should be suitable VR devices to improve cognitive function.

A previous systematic review and meta-analysis suggested that sufficient cognitive challenges seemed important for a combined program of physical and cognitive activity [38]. In our systematic review, we focused on the gaming factors which could enhance motivation rather than cognitive challenges of PE programs combined with VR devices. Not only the studies used VR-PE with cognitive challenges [17,18,19,20,23,24,25] but also without high cognitive challenges [16,21,22,26] reported significant improvement of cognitive functions. In future studies, it would be important to investigate the effect of cognitive or psychological components of the VR-PE such as cognitive challenges and gaming factors on improving cognitive function.

A previous meta-analytical review suggested that balance training using VR was an acceptable method for improving balance performance and functional mobility in community-dwelling older adults [39]. In this SR, three studies were performed for community-dwelling older adults with and without mild cognitive decline [17,18,26]. Two studies of healthy older adults reported a significant improvement compared to the control group [26] and pre-post change [18] in general cognitive function. Moreover, one study performed for older adults with mild cognitive decline reported significant improvements in long-term memory and a self-report measure of cognitive function, and a pre-post difference in executive function [17]. Although the number of included studies was small, this SR suggested the potential of the VR-PE to improve cognitive functions of community-dwelling older adults.

This study had some limitations. The first limitation was the small number of included studies. We found only six studies in healthy older adults and five studies in older adults with mild cognitive decline investigating VR effects on physical exercise. The second limitation is that previous studies used a wide variety of cognitive function measures, only a few studies used common cognitive function measures, such as the Stroop task. The third limitation was the small sample size of the included studies. Four of the eleven studies included fewer than 10 participants in each group [17,19,23,25]. Due to these limitations, it is difficult to generalize the beneficial effects of VR-PE on each cognitive domain. However, as discussed above, the available evidence was enough to show that VR-PE has the potential to improve cognitive function in older adults.

## 5. Conclusions

We first conducted the SR for RCTs to investigate the benefits of VR-PE on cognitive function in older adults with and without mild cognitive decline. From 11 RCT studies (six studies for healthy older adults and five studies for older adults with mild cognitive decline), six studies reported significant improvements in several cognitive functions (general cognitive function, executive functions, working memory capacity, short-term memory, and verbal episodic memory) compared to the control group. The SR revealed that VR-PE could have beneficial effects on inhibition in older adults with and without mild cognitive decline. Moreover, VR-PE selectively leads to improvements in shifting and general cognitive performance in healthy older adults. The SR indicates that VR-PE would have a possibility to improve cognitive function in older adults. However, due to a small number of included studies, further studies will be necessary to draw a conclusion of the benefits of VR-PE on cognition. In addition, it would be important to investigate whether the VR-PE would have a positive effect on cognition in the young population as well as an aging population.

## Figures and Tables

**Figure 1 healthcare-09-00883-f001:**
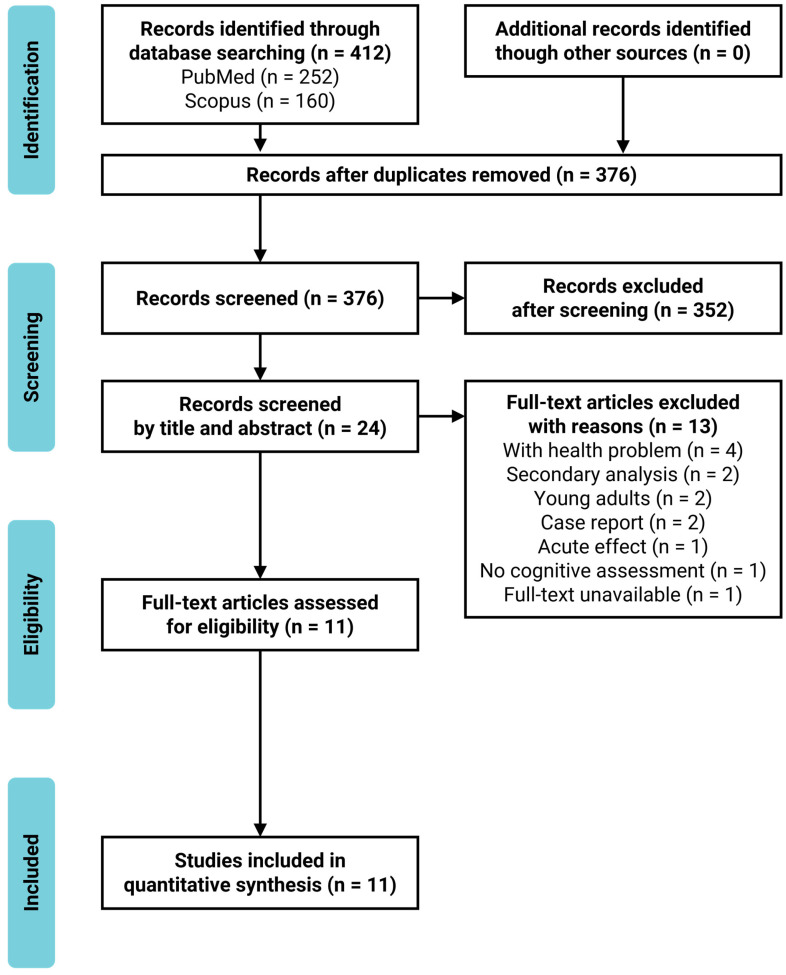
PRISMA flowchart.

**Table 1 healthcare-09-00883-t001:** Characteristics of participants in the included studies.

Lead Author; Year; Country	Sample Size (Female (%))	Age	Cognitive Status
Hsieh; 2018; Taiwan *	60 (72%)	(mean ± SD)	MMSE score 11–26
	VE: 31	VE: 76.4 ± 7.6	
	NC: 29	NC: 80.0 ± 7.5	
Eggenberger; 2015; Switzerland †	71 (65%)	(mean ± SD)	MMSE (mean ± SD)
	VE1: 24	VE1: 77.3 ± 6.3	VE1: 28.4 ± 1.4
	VE2: 22	VE2: 78.5 ± 5.1	VE2: 28.3 ± 1.2
	EC: 25	EC: 80.8 ± 4.7	EC: 28.0 ± 1.7
Anderson-Hanley; 2018; United States *	14 (50%)	(mean ± SD)	MoCA (mean ± SD)
	VE: 7	VE: 75.4 ± 9.83	VE: 22.0 ± 3.21
	EC: 7	EC: 80.9 ± 12.3	EC: 21.6 ± 2.70
Mrakic-Sposta; 2018; Italy (n.s.)	10 (60%)	(mean ± SD)	MMSE (mean ± SD) 23.0 ± 3.4
	VE: 5	VE: 72.0 ± 5.15	
	NC: 5	NC: 74.6 ± 6.43	
Anderson-Hanley; 2012; United States *	79 (78%)	(mean ± SD)	Performance ≤ −1.5 SD of norm on at least one subtest (n)
	VE: 38	VE: 75.7 ± 9.9	VE: 16
	EC: 41	EC: 81.6 ± 6.2	EC: 14
Bacha; 2018; Brazil †	46 (74%)	(Medium [Q1; Q3])	MoCA (mean ± SD)
	VE: 23	VE: 71.0 (66.0; 74.5)	VE: 23.48 ± 4.94
	EC: 23	EC: 66.5 (65.0; 71.8)	EC: 22.52 ± 3.47
Htut; 2018; Thailand *	84 (44%)	(mean ± SD)	MMSE (mean ± SD)
	VE: 21	VE: 75.8 ± 4.89	VE: 25.5 ± 1.22
	EC: 21	EC: 75.9 ± 5.65	EC: 24.7 ± 0.96
	CC: 21	CC: 75.6 ± 5.33	CC: 25.2 ± 1.41
	NC: 21	NC: 76.0 ± 5.22	NC: 25.2 ± 1.00
Delbroek; 2017; Belgium (n.s.)	20 (65%)	(mean ± SD)	MoCA (mean ± SD)
	VE: 10 (dropout: 2)	VE: 86.9 ± 5.6	VE: 17.7 ± 5.3
	NC: 10 (dropout: 1)	NC: 87.5 ± 6.6	NC: 16.8 ± 5.8
Liao; 2020; Taiwan †	34 (68%)	(mean ± SD)	MMSE (mean ± SD)
	VE: 18	VE: 75.5 ± 5.2	VE: 27.2 ± 1.9
	ECC: 16	ECC: 73.1 ± 6.8	ECC: 28.3 ± 1.2
Monteiro-Junior; 2017; Brazil *	18 (67%)	(mean ± SD)	MMSE (mean ± SD)
	VE: 9	VE: 85.0 ± 8.0	VE: 21.0 ± 5.0
	EC: 9	EC: 86.0 ± 5.0	EC: 24.0 ± 4.0
Park; 2016; Korea *	72 (94%)	(mean ± SD)	MoCA (mean ± SD)
	VE: 36	VE: 73.0 ± 3.0	VE: 22.6 ± 4.9
	EC: 36	EC: 74.1 ± 2.9	EC: 22.9 ± 4.2

Note. VE: VR-based exercise group; EC: exercise only control group; CC: cognitive only control group; ECC: exercise and cognitive control group; NC: no treatment control group; SD: standard deviation; Q1: first quartile; Q3: third quartile; MMSE: mini-mental state examination; MoCA: Montreal cognitive assessment; *: statistically significant improvement compared to the control group; †: statistically significant improvement from the baseline; n.s.: not significant.

**Table 2 healthcare-09-00883-t002:** Quality assessment scores of included studies using modified Delphi list.

Lead Author; Year; Country	Q1	Q2	Q3	Q4	Q5	Q6	Q7	Q8	Q9	Q10	Q11	Q12	Q13	Q14	Total Score (Max. = 14)
Hsieh; 2018; Taiwan	N	?	Y	Y	N	N	N	Y	Y	N	N	Y	Y	N	6
Eggenberger; 2015; Switzerland	Y	Y	Y	Y	N	N	Y	N	N	Y	Y	Y	Y	Y	10
Anderson-Hanley; 2018; United States	Y	?	Y	Y	?	?	?	Y	N	N	Y	Y	Y	Y	8
Mrakic-Sposta; 2018; Italy	Y	?	Y	Y	?	?	?	Y	N	N	N	Y	Y	N	6
Anderson-Hanley; 2012; United States	Y	Y	Y	Y	Y	?	?	Y	Y	Y	Y	Y	Y	Y	12
Bacha; 2018; Brazil	Y	Y	Y	Y	Y	?	N	Y	N	Y	Y	Y	Y	Y	11
Htut; 2018; Thailand	Y	?	Y	Y	Y	?	N	N	N	N	Y	Y	Y	N	7
Delbroek; 2017; Belgium	Y	?	Y	Y	Y	?	?	Y	N	N	N	N	Y	N	6
Liao; 2020; Taiwan	Y	Y	Y	Y	Y	N	N	Y	N	N	Y	Y	Y	N	9
Monteiro-Junior; 2017; Brazil	Y	?	Y	Y	?	N	Y	N	N	N	Y	Y	Y	N	7
Park; 2016; Korea	Y	Y	Y	Y	?	?	?	Y	N	N	Y	Y	N	N	7
Total score across studies	10	5	11	11	5	0	2	8	2	3	8	10	10	4	-
Average of total score ± SD															8.1 ± 2.02

Note. Q1: Random allocation; Q2: Treatment allocation concealed; Q3: Groups/subjects similar at baseline regarding important prognostic values; Q4: Eligibility criteria specified; Q5: Blinded outcome assessor; Q6: Care provider blinded; Q7: Patient blinded; Q8: Point estimates and measures of variability presented for the primary outcome measures; Q9: Intention-to-treat analysis; Q10: Details of random allocation methods; Q11: Adequate description of the control/comparison group; Q12: Between-group statistical comparison; Q13: Reporting dropout; Q14: Reporting CONSORT statement; Y: Yes; the study met the criteria of the question; N: No; the study did not meet the criteria of the question; ?: No information or the study was not related to the question; SD: Standard deviation.

**Table 3 healthcare-09-00883-t003:** Characteristics of intervention methods in the included studies.

Lead Author; Year; Country	Periods of Intervention	VR Device	Experimental Group	Control Group
Hsieh; 2018; Taiwan *	Two 60 min sessions/week for 6 months	Xbox 360 Kinect, 100-inch flat screen	VE: VR-based Tai Chi exercise (Chinese mind-body exercise with biofeedback)	NC: No intervention
Eggenberger; 2015; Switzerland †	Two 60 min sessions/week for 6 months	Impact Dance Platform, large screen	VE1: Video game dancing on the pressure sensitive areas to detect steps VE2: Treadmill walking with verbal memory exercise	EC: Treadmill walking without an additional cognitive task
Anderson-Hanley; 2018; United States *	≥Two ≥20 min sessions/week for 6 months	Cybercycle, virtual reality display (small monitor)	VE: VR-bike rides with effortful cognitive videogame	EC: VR-bike rides with passive cognitive processing
Mrakic-Sposta; 2018; Italy (n.s.)	Three 40–45 min sessions/week for 6 weeks	Cycle-ergometer, finger touch projector	VE: Cycling exercise with VR-based cognitive training	NC: No treatment
Anderson-Hanley; 2012; United States *	Five 45 min sessions/week for 3 months	Cybercycle, virtual reality display (small monitor)	VE: VR-bike rides with virtual 3D tours	EC: Traditional stationary bike rides
Bacha; 2018; Brazil †	Two 60 min sessions/week for 7 weeks	Xbox Kinect Adventures, 50-inch TV	VE: Playing Kinect games including cognitive demands	EC: Conventional physical therapy exercises
Htut; 2018; Thailand *	Three 30 min sessions/week for 8 weeks	Xbox 360	VE: VR-based exercise games	EC: Strength and balance exercises CC: Brain exercise games NC: no exercise
Delbroek; 2017; Belgium (n.s.)	Two 18–30 min sessions/week for 6 weeks	BioRescue, 55-inch TV-flat screen	VE: VR cognitive-motor dual task training	NC: No intervention (usual care)
Liao; 2020; Taiwan †	Three 60 min sessions/week for 12 weeks	Microsoft Kinect, VIVE system	VE: VR-based physical and cognitive training	ECC: Combined physical and cognitive training
Monteiro-Junior; 2017; Brazil *	Two 30–45 min sessions/week for 6–8 weeks	Wii Fit Plus	VE: Playing VR-based exercise games	EC: Playing exercise games
Park; 2016; Korea *	Two 30+20 min sessions/week for 6 weeks	Kayak ergometers 3-D beam projector	VE: Conventional exercise & VR kayak paddling exercise	EC: Conventional exercise

Note. VE: VR-based exercise group; EC: Exercise only control group; CC: Cognitive only control group; ECC: Exercise and cognitive control group; NC: No treatment control group; *: Statistically significant improvement compared to the control group; †: Statistically significant improvement from the baseline; n.s.: Not significant.

**Table 4 healthcare-09-00883-t004:** Description of Measured Cognitive Function.

Lead Author; Year; Country	Measured Cognitive Functions								Statistical Method
	General Cognitive Function	Executive Function	Working Memory	Short-Term Memory	Episodic Memory	Processing Speed	Attention	Others	
Hsieh; 2018; Taiwan		ABSTR in CASI *, ANML in CASI (n.s.)		short-term memory in CASI (n.s.)			ATTEN in CASI (n.s.)	subtests in CASI (MENMA, orientation, language, drawing, long-term memory) (n.s.)	generalized estimating equation (GEE) analyses
Eggenberger; 2015; Switzerland		TMT-B †, executive control task †	digit backward tasks in WMS-R †	digit forward tasks in WMS-R (n.s.)	logical memory subtest (story recall) in WMS-R †, paired-associates learning task †	TMT-A †, DSST in WAIS-R †	age concentration tests A and B †		multiple regression analysis
Anderson-Hanley; 2018; United States		Stroop †, color trails (n.s.)	digit span (n.s.)		ADAS word list (immediate recall, delayed recall) *			ecological validity *	repeated measures ANCOVA
Mrakic-Sposta; 2018; Italy	MMSE (n.s.)	FAB (n.s.), TMT-B (exculded), VF (n.s.)			RAVLT_I (n.s.), RAVLT_D (n.s.)	TMT-A (n.s)	AM (n.s.)	ROCFT (n.s.), FAQ (n.s.)	Mann-Whitney U-Test
Anderson-Hanley; 2012; United States		color trails *, Stroop C *, COWAT (n.s.), categories (n.s.)	digit span backwards *		RAVLT_I (n.s.), RAVLT_D (n.s.), Fuld delayed recall (n.s.), figure delayed recall (n.s.)		LDST (n.s.)	figure copy (n.s.), clock (n.s.)	repeated measures ANCOVA
Bacha; 2018; Brazil	MoCA †								repeated measures ANOVA
Htut; 2018; Thailand	MoCA *	TUG-cog *							two-way mixed ANOVA
Delbroek; 2017; Belgium	MoCA (n.s.)	TUG-DT (n.s.)							Wilcoxon signed-rank test two-sided
Liao; 2020; Taiwan †	MoCA †	EXIT-25 †			CVVLT (immediate recall, delayed recall) †				repeated measures ANOVA
Monteiro-Junior; 2017; Brazil *	MMSE (n.s.)	FMT (route, recall) (n.s.), VF(n.s.)	digit span backward (n.s.)	digit span forward *		TMT-A (n.s.)			independent *t* tests
Park; 2016; Korea *	MoCA *								Independent *t* tests

Note. *: Statistically significant improvement compared to the control group; †: Statistically significant improvement from the baseline; n.s.: Not significant; MMSE: Mini-mental state examination; MoCA: Montreal cognitive assessment; ABSTR: Abstract thinking and judgment; CASI: Cognitive abilities screening instrument; ANML: Animal name fluency; ATTEN: Attention; MENMA: Mental manipulation; TMT-B: Trail making test part B; WMS-R: Wechsler memory scale-revised; TMT-A: Trail making test Part A; DSST: Digit symbol substitution task; WAIS-R: Wechsler adult intelligence scale-revised; ADAS: Alzheimer’s disease assessment scale; FAB: Frontal assessment battery; VF: Verbal fluency test; RAVLT_I: Immediate recall of Rey auditory verbal learning test; RAVLT_D: Delayed recall of Rey auditory verbal learning test; AM: Attentional matrices test; ROCFT: Rey–Osterrieth complex figure test; FAQ: Functional activity questionnaire; COWAT: Controlled oral word association test; LDST: Letter digit symbol test; TUG-cog: Timed up and go test cognition; TUG-DT: Timed up and go test dual attention task; EXIT-25: Executive interview 25; CVVLT: Chinese version of the California verbal learning test; FMT: Floor maze test; ANOVA: Analysis of variance; ANCOVA: Analysis of covariance.

## Data Availability

Not applicable.

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
