# Peer review of "Benefits of VR Physical Exercise on Cognition in Older Adults with and without Mild Cognitive Decline: A Systematic Review of Randomized Controlled Trials"

_healthcare, 2021, doi:10.3390/healthcare9070883_

Round 1

Reviewer 1 Report

General comments

The present systematic review aimed to investigate whether physical exercise combined with virtual reality would be beneficial for cognitive functions in older adults. The study covers a very hot topic, given recent evidence supporting the fundamental role of exercise and physical activity for improving cognitive functions in adults (or preventing cognitive decline). The manuscript is well-written, easy to read, and provides new insights into the potential benefit of exercise with virtual reality on cognition in older adults. Procedures are well described and rigorously conducted. I would congratulate the Authors for their work.

Although I strongly believe that the manuscript has merit and its contents should be shared with the scientific community, apart from a couple of minor comments (see below), there is an issue that should be considered. A similar systematic review and meta-analysis has been recently published (Gheysen et al., 2018). This study aimed to summarize the literature on the potential positive effects of physical activity programs enriched with cognitive challenges on cognition in older adults. I would suggest to refer to this recent study, highlighting the similarities and differences with the study by Gheysen and colleagues. Although virtual reality is not the same as cognitive challenges, the underlying effects may be partially overlapped, and this should be considered when discussing the findings of the present systematic review.  

Specific comments

Line 38-39. I would suggest to extend this part by mentioning that various types of exercises have been shown to (both acutely and chronically) improve cognitive functions in middle-aged and older adults. Here some suggestions to cite:

Formenti, D., Cavaggioni, L., Duca, M., Trecroci, A., Rapelli, M., Alberti, G., Komar, J., Iodice, P., 2020. Acute Effect of Exercise on Cognitive Performance in Middle-Aged Adults: Aerobic Versus Balance. J. Phys. Act. Health 17, 773–780. https://doi.org/10.1123/jpah.2020-0005

Hsu, C.L., Best, J.R., Davis, J.C., Nagamatsu, L.S., Wang, S., Boyd, L.A., Hsiung, G.R., Voss, M.W., Eng, J.J., Liu-Ambrose, T., 2018. Aerobic exercise promotes executive functions and impacts functional neural activity among older adults with vascular cognitive impairment. Br. J. Sports Med. 52, 184–191. https://doi.org/10.1136/bjsports-2016-096846

Voelcker-Rehage, C., Godde, B., Staudinger, U.M., 2011. Cardiovascular and Coordination Training Differentially Improve Cognitive Performance and Neural Processing in Older Adults. Front. Hum. Neurosci. 5. https://doi.org/10.3389/fnhum.2011.00026

Discussion-conclusion. Please give space and discuss possible practical applications derived from the evidence of this study. Do the Authors consider that virtual reality-physical exercise should be implemented with specific policies? What about community-dwelling people and the others?

Reviewer 2 Report

The authors have conducted a systematic review of research on the efficacy of virtual reality-assisted physical exercise on cognitive function in older adults. The authors list several rather severe limitations in their discussion, including limited studies (n=8), limited studies including adults with and without mild cognitive decline (n=4), extremely small sample size in some studies (smallest N = 5 per group), variety of methodologies, and variety of outcome measures. The authors themselves state, “Due to these limitations, it is difficult to generalize the beneficial effects of VR-PE on each cognitive domain.” And yet the authors then go on to conclude that VR-PE has great potential. However, given the limitations the authors list and acknowledge, and given the mixed findings (only half the studies found significant results), it appears that there simply is not enough data support drawing any conclusion. With so few studies, it is simply premature to attempt a systemic review at this time.

Additional Concerns

  1. “This decline is an indicator of lower well-32 being [2] and dementia in the future [3, 4].” This sentence implies that all older adults will experience dementia, which is not true.

  1. Grammatical correction: “in which people at their homes USE commercially available VR systems”

  1. It is unclear why Table 1 is the first table, it seems that it would be better positioned after the current Table 2, with section 3.2 of the paper. Lines to 125 to 134 completely duplicate information already presented in Table 1 and are therefore unnecessary.

  1. It is unclear what is defined as “sufficient methodological quality” – is there a specific score that denotes sufficiency?

  1. A table indicating significant and nonsignificant results across studies/measures would provide a very helpful summary of the SR. It would helpful to know quickly which studies reported in Table 2 found significant results: to link methodologies and sample sizes with outcomes.

  1. The discussion section is mostly a repeat of the results section and offers little interpretation of the results of the SR.

Reviewer 3 Report

The paper is well structured, well written and clear in the presentation of the work performed and the obtained results.

This research contributes to gather and analyze existing evidence on the benefits of virtual reality-based exercise on the cognition of older adults (healthy or with mild cognitive decline).

The main limitation is the reduced number of primary sources that were selected after the application of exclusion criteria. Many constraints were imposed on selected papers, leading to the exclusion of potentially related studies. Considering only older adults (older than 60) without any neurological disorder excludes research that has also found benefits of VR-based exercise on cognition in this segment of older adults. RCTs that investigated this potential benefit with younger people are also excluded.

As a result, the systematic review is very well focused on a specific research question, but the conclusions are supported by limited evidence. This limitation is recognized by the authors.

I also wonder if more papers could have been found by searching in additional databases. It seems that the queries were only run in PubMed. Given that virtual reality researchers tend to publish in venues that might not be indexed in PubMed, I would suggest to expand the search to wider-scope databases such as Scopus. 

Minor errors:

  • Section 3.1 states that the sample size ranged from 10 to 111. According to the data in Table 2, the maximum sample size is 84.
  • Section 3.3 states that the intervention period ranged from 4 weeks to 6 months. According to data in Table 3, the minimum intervention period is 6 weeks.
  • Last paragraph of section 4: the next statement is incorrect:

    We found only four studies in older adults with and without mild cognitive decline investigating VR effects on physical education. 
